# Maternal Underestimation of Child’s Weight at Pre-School Age and Weight Development between Age 5 and 12 Years: The ABCD-Study

**DOI:** 10.3390/ijerph17145197

**Published:** 2020-07-18

**Authors:** Tanja G. M. Vrijkotte, Tina M. C. K. Varkevisser, Daniel B. van Schalkwijk, Marieke A. Hartman

**Affiliations:** 1Department of Public and Occupational Health, Amsterdam Public Health Research Institute, Amsterdam UMC, University of Amsterdam, 1105 AZ Amsterdam, The Netherlands; tina.varkevisser@gmail.com (T.M.C.K.V.); m.a.hartman@euc.eur.nl (M.A.H.); 2Department of Sciences, Amsterdam University College, VU Amsterdam/University of Amsterdam, 1012 WX Amsterdam, The Netherlands; D.B.vanSchalkwijk@auc.nl; 3Department of Life Sciences, Erasmus University College, Erasmus University Rotterdam, 3062 PA Rotterdam, The Netherlands

**Keywords:** overweight, maternal perception, pre-school age, BMI development, ethnicity, paternal BMI, maternal BMI

## Abstract

Background: Healthcare monitoring of child growth reduces with age, which may increase parental influences on children’s weight development. This study aimed to examine the association between maternal underestimation of child’s weight at age 5/6 and weight development between 5 and 12 years. Methods: We performed univariate and multivariate linear regression analyses with data on maternal perception of child’s weight and weight development (∆SDS body-mass index; BMI) derived from the Amsterdam Born Children and their Development (ABCD) birth-cohort study. Underestimation was defined by comparing maternal perception of child’s weight with the actual weight status of her child. Associations were studied in two groups: children with overweight (*n* = 207) and children with normal weight (*n* = 1982) at baseline (children with underweight were excluded). Results: Underestimation was 5.5% in children with normal weight and 79.7% in children with overweight. Univariate analyses in children with normal weight and overweight showed higher weight development for children with underestimated vs. accurately estimated weights (respectively: β = 0.19, *p* < 0.01; β = 0.22, *p* < 0.05). After adjusting for child sex and baseline SDS BMI, the effect size became smaller for children with a normal weight (β = 0.15, *p* < 0.05) and overweight (β = 0.18, *p* > 0.05). Paternal and maternal BMI, ethnicity, and educational level explained the association further (remaining β = −0.11, *p* > 0.05 in children with normal weight; β = 0.06, *p* > 0.05 in children with overweight). Conclusions: The relationship between maternal underestimation of child’s weight and higher weight development indicates a need for promoting a realistic perception of child’s weight, this is also the case if the child has a normal weight.

## 1. Introduction

The prevalence of childhood overweight and obesity has been increasing worldwide, including in the Netherlands [1]. In the Netherlands, one in five children are overweight, and this increases with age [1,2]. Subsequently, childhood overweight commonly tracks into adulthood [3,4]. Given the increasing prevalence of overweight and the associated comorbidities and complications, both long and short-term, the need for prevention and treatment becomes urgent [5]. In the Netherlands, a youth healthcare system monitors child development to prevent unhealthy weight gain by providing health check-ups [6]. After the age of 2, however, check-ups become less frequent, shifting a greater responsibility of child’s weight to parents [6]. Parental perception of child’s body weight may therefore be an important influence on children’s healthy weight development. Current literature shows that mothers commonly misperceive their child’s weight [7,8,9,10,11,12,13,14,15,16]. However, it remains unclear what the long-term effect is of maternal underestimation of their child’s weight on the child’s weight development. To the best of our knowledge, only three studies to date have looked at this association longitudinally and found contradictory results. In a German study, maternal underestimation was associated with increased body mass index (BMI), while in two other studies that were conducted in the Netherlands and Peru, maternal underestimation was associated with greater BMI loss [8,11,15].

Demographic variables may explain differences in the association of underestimation of child’s weight and child’s weight development. Ethnicity is an example that has been found to have an influence. A study from the Netherlands found that maternal underestimation of child’s weight was more common among Turkish and Moroccan parents compared to ethnically Dutch parents [13]. Maternal education level has also been found to largely explain maternal underestimation of child’s weight [7,8,11,12,13,15,16]. Overall, studies suggest that certain characteristics such as parental age, educational level, parental weight status, ethnicity and child sex are associated with maternal misperception of child’s weight [7,8,11,13,15,16] as well as the risk of a child becoming overweight [7,13,16]. Understanding the explanatory role of these demographic characteristics in the association of underestimation and child development could help identify certain risk groups.

To better understand the contradictory results on the relationship between maternal underestimation of child’s weight and child’s weight development, this paper examines the association between maternal perception of child’s weight at age five in normal weight and overweight children from Amsterdam, the Netherlands, and child’s weight development between 5/6 and 11/12 years of age. Additionally, this paper explores which demographic factors may explain differences in this association. This is the first paper to study the association between maternal underestimation and child’s weight development in a multi-ethnic Dutch cohort, with a long follow-up time and large sample size. We hypothesize that children whose weight is underestimated by their mothers at the age of 5/6 will have a higher weight development (change in standardized body mass index (SDS BMI) in six years, compared to children whose weight is accurately estimated. We expect to see this positive weight development for children with normal weight at baseline as well as those with overweight. Maternal underestimation of those with normal weight would imply that mothers believe their children are underweight, which may encourage unhealthy weight gain, while maternal underestimation of those with overweight implies that mothers believe their children’s weights are just right, which may not encourage weight loss. Finally, we hypothesize that the higher weight development in underestimated children can be explained by higher maternal and paternal BMI, lower maternal educational level, and non-Dutch ethnicity.

## 2. Materials and Methods

### 2.1. Study Design and Data Source

We used data from the Amsterdam Born Children and their Development (ABCD) study. The ABCD study is a long-term, large-scale, birth cohort study, which aims to examine and determine factors in early life that may explain the health of children at a later stage. Extensive information on the ABCD study is available elsewhere [17]. The study is conducted according to the Declaration of Helsinki and approval was obtained from the Academic Medical Center Medical Ethics Committee (MEC 02/039#02.17.392), the VU University Medical Center Medical Ethical Committee, and the Registration Committee of Amsterdam [17]. All participating mothers provided written informed consent for themselves and their children. Twelve-year-old children also gave informed consent for themselves.

The ABCD study started in 2003 when expecting mothers living in Amsterdam were invited to participate. In this study on the association of maternal perception with child’s weight development, data collected from the ABCD cohort in 2008–2010 and 2015–2016 were used, when the children were 5–6 years and 11–12 years of age.

### 2.2. Participants

In 2008, 6161 mothers gave permission for follow-up and received a questionnaire, of which 4488 were returned (73%). Reasons for participants lost to follow-up included withdrawal from the study, infant or maternal death, unknown addresses, or emigration.

We excluded participants with missing data on child age, sex, BMI, and maternal perception (*n* = 2005), as these were crucial variables for our analysis. Underweight children (*n* = 294) were also excluded because the study was interested in weight development of children who were initially normal weight or overweight. After exclusion, the study included 2189 participants with BMI data at age 5–6 as well as at age 11–12. 

### 2.3. Dependent Variable: ∆ SDS BMI between 5/6 and 11/12 Years 

The dependent variable was the change in age- and sex- standardized BMI (∆ SDS BMI) between 5/6 and 11/12 years of age, referred to as child’s weight development. Data on weight and height were derived from the ABCD health checks at age 5/6 and 11/12, and the Amsterdam Youth Health care registration to calculate the children’s BMI. Trained personnel measured child’s height and weight according to standard procedures [13]. Subsequently, SDS BMI was calculated at both ages by GrowthAnalyser using the WHO growth reference [18]. Weight development was determined by subtracting SDS BMI at age 5/6 from SDS BMI at age 11/12. 

### 2.4. Independent Variable: Maternal Perception of Child’s Weight

The independent variable was the mother’s ability to perceive her child’s weight accurately (i.e., maternal perception). To define this variable, we compared the maternal perception data at age 5/6 with the actual weight status of the children at age 5/6. The children’s weight statuses were determined by the International Obesity Task Force (IOTF) guidelines, using age-and sex-specific BMI cut off points for three categories: underweight, normal weight, and overweight (including obesity) [19]. Maternal perception was measured through questionnaires in which mothers were asked what they thought of their child’s weight status based on a 5-point scale (1 = way too heavy; 2 = too heavy; 3 = just right; 4 = too light; 5 = way too light). If mothers perceived their child to be a lower weight category than they actually were (e.g., a child with overweight being perceived as normal weight) we called this underestimation. On the other hand, if mothers perceived their child’s weight accurately, in the correct weight group, we called this accurate estimation [13].

### 2.5. Other Variables

The additional variables that were studied included: child age at baseline and follow-up measurement, child sex, ethnicity, maternal age during pregnancy (self-reported), maternal BMI (self-reported height and weight), paternal BMI (reported by mother), and maternal education level at child’s age 5/6. Ethnicity was determined by the mother’s country of birth and was classified as “Dutch”, “Surinamese”, “Turkish”, “Moroccan” or “Other”. Maternal education level was classified as: “low”, if she had completed no education or primary school, lower vocational secondary education, technical secondary education, or less than 6 years after primary school; “middle”, if she had completed higher vocational secondary education, intermediate vocational education, or 6–10 years of education after primary school; or “high”, after obtaining higher vocational education, university education, or more than 10 years of education after primary school [20].

### 2.6. Statistical Analyses

Statistical analyses were conducted using the IBM SPSS package version 23.0 (SPSS Inc., Armonk, NY, USA). We considered *p*-values <0.05 as statistically significant. Demographic variables were described (mean, SD, and proportions) according to the weight status of the child at age 5/6 and maternal child’s weight perception. Differences in demographic variables were tested using 𝛘^2^-square tests for categorical variables, and T-tests for continuous variables. Linear Regression Models were used to explore the association between maternal child’s weight perception and child’s weight development. Analyses were done separately for children with a normal weight and children with overweight at age 5/6. First, a univariate analysis was performed to study the association between maternal child’s weight perception and child’s weight development. Then, two multivariate models were built: the first adjusted for child sex and baseline SDS-BMI, to adjust for variation in SDS BMI within the spectrum of normal weight and overweight, respectively (Model 1), and the second adjusted further for maternal and paternal BMI, maternal age, ethnicity, and educational level (Model 2) to investigate the explanatory role of these characteristics on the association between maternal perception and child’s weight development. Growth differs between boys and girls [21] and some studies show sex differences to be in agreement between measured and perceived child’s weight by mothers [22]. Therefore, effect modification by child’s sex in the association between maternal child’s weight perception and weight development was tested by creating an interaction term between maternal child’s weight perception and child’s sex and adding it to the multivariable regression analysis (Model 2). If the added interaction coefficient had a *p*-value of <0.10, effect modification was assumed.

## 3. Results

### 3.1. Study Population

The prevalence of overweight was 7.9% at baseline and 10.8% at follow-up. The prevalence of obesity was 1.6% and 1.8%, respectively. Henceforth, overweight and obesity are grouped together and referred to as “children with overweight”. In the group of children with normal weight, 5.5% of the mothers underestimated their child’s weight, while 79.7% of the mothers of overweight children underestimated their child’s weight.

Table 1 shows the characteristics of the 2189 participants of this study according to child’s weight status and maternal child’s weight perception at baseline. Children with a normal weight (*n* = 1982) tended to have parents with a lower BMI (*p* < 0.001), mothers who were older (*p* < 0.001, were more educated (*p* < 0.001), and were ethnically Dutch (*p* < 0.001) compared to children with overweight (*n* = 207). The children with normal weight were also younger on average during the baseline measurement (*p* < 0.001), older during the follow-up measurement (*p* < 0.001), and comprised of more boys than girls (*p* < 0.01) compared to the children with overweight.

Compared to children whose weight was underestimated by their mother (*n* = 275), children whose weight was accurately estimated (*n* = 1914) had older mothers (*p* < 0.001), with a lower BMI (*p* < 0.001), who were higher educated (*p* < 0.001), and more often ethnically Dutch (*p* < 0.001). The children whose weight was accurately estimated were also younger on average at baseline (*p* < 0.001), older at follow-up (*p* < 0.001), and had a lower baseline SDS BMI compared to the children whose weight was underestimated (*p* < 0.001).

### 3.2. Maternal Perception and Weight Development

#### 3.2.1. Children with a Normal Weight

Univariate analyses showed that weight development (i.e., ∆ SDS BMI) between ages 5/6 and 11/12 was significantly higher (β = 0.19;95%CI: 0.05, 0.33) in children with normal weight whose mothers underestimated their weight, compared to children whose mothers accurately estimated their weight (Table 2) (*p* < 0.01). The effect size was slightly reduced, but remained statistically significant (β = 0.15; 95%CI: 0.01, 0.30) after minimal adjustments (Table 2, Model 1). After adjusting further for demographic characteristics (Table 2, Model 2), the effect sizes decreased (β = −0.11; 95%CI: −0.28, 0.05) resulting in loss of significance. Child sex (β = 0.15; 95%CI: 0.09, 0.22), baseline SDS BMI (β = −0.17; 95%CI: −0.23, −0.11), maternal BMI (β = 0.03; 95%CI: 0.02, 0.04), and paternal BMI (β = 0.03; 95%CI: 0.02, 0.04) were all independently associated with weight development. Moreover, compared to ethnically Dutch children, Turkish children and children of other ethnicities showed increased weight development. Children of low and middle educated mothers also showed increased weight development compared to children of highly educated mothers.

Offspring’s sex did not act as an effect modifier, the *p*-value for interaction was 0.426.

#### 3.2.2. Children with Overweight

Univariate analyses in the children with overweight showed that weight development between ages 5/6 and 11/12 was significantly higher (β = 0.22; 95%CI: 0.01, 0.44) in children whose mothers underestimated their weight, compared to children whose mothers accurately estimated their weight (Table 2) (*p* < 0.05). After minimal adjustments (Table 2, Model 1), the effect size decreased (β = 0.18; 95%CI: −0.04, 0.41), resulting in the loss of significance in the differences in maternal perception. Further adjustments for demographics (Table 2, Model 2) again showed a decreased effect size (β = 0.06; 95%CI: −0.20, 0.31. Only baseline SDS BMI (β = −0.34; 95%CI: −0.56, −0.11) and maternal BMI (β = 0.05; 95%CI: 0.03, 0.07) were independently associated with weight development in children with overweight.

Offspring’s sex did not act as an effect modifier, the *p*-value for interaction was 0.638.

## 4. Discussion

This study explored the association between maternal perception of child’s weight and the child’s weight development measured as ∆ SDS BMI between the age of 5/6 and 11/12. Analyses showed that children with a normal weight whose mothers underestimated their weight at age 5/6 displayed a higher weight development in the six years that followed. The same trend was seen for children with overweight, although it was not statistically significant when adjusted for child sex and baseline SDS BMI. Higher maternal or paternal BMI, being from Turkish or other non-Dutch ethnicity, and lower maternal educational level were all independently associated with increased weight development in children with normal weight. These demographic characteristics explained the difference in weight development between children whose body weight was underestimated and accurately estimated.

### 4.1. Comparison with Other Studies

In comparison to findings from previous longitudinal studies, our study explains the association of maternal perception of child’s weight and child’s weight development in children. Similar to our results, a study in Germany reported that children with overweight whose weight was underestimated in early development had a more pronounced weight gain in the future compared to children who were considered to be just right [15]. In contrast, a Dutch study found that children with overweight whose weight was underestimated had a greater SDS BMI loss compared to children whose overweight was accurately estimated [11]. The weight of these mainly ethnically Dutch children was on average normal at the time before and after parental perception of child’s weight was measured. This might explain why parents were more likely to perceive their child as normal weight, and why these children experienced more, rather than less, weight loss compared to children whose weight was accurately perceived. Unfortunately, both studies did not investigate the longitudinal BMI change in children with normal weight as a function of maternal perception.

Differences in findings between our study and other longitudinal studies might be further explained by socio-cultural factors. We found an association between non-Dutch ethnicity and lower maternal education, with maternal underestimation and higher weight development in children with a normal weight. In accordance, ethnic and cultural differences have been found for body image acceptance, accurate body weight perception, ideal body size perception, and weight loss behavior [13,23,24,25,26,27,28,29]. Studies on the perception of body weight among adults showed that overweight men/women from ethnic minorities were more likely to perceive their own weight as normal [26,30]. The greater acceptance of larger body size in some cultures may account for these differences [26,30]. Because of a higher prevalence of overweight in some ethnic groups, overweight is more common in the familial and social environment [31]. Furthermore, in some cultures mothers do not associate childhood obesity with health, in fact it is regarded as a sign of wealth and thinness may be more strongly associated with poor health [28,29]. Moreover, maternal education level is an important factor in the mother’s ability to accurately perceive her child’s weight [7,8,11,12,13,15,16]. Additionally, cross-national differences may apply in the relationship between maternal underestimation of child’s weight and child’s weight development. In contrast to developed countries such as the Netherlands and Germany, in Peru, having a high SES and living in Lima increases the chance of developing overweight and obesity as a child [32]. Carillo-Larco et al. included a study sample from resource-limited urban areas as well as rural areas in Peru, which suggests that resources to allow weight gain were limited too. This may explain why Carillo-Larco et al. found a greater loss in children’s BMI whose weight was underestimated [8], while our Dutch study and Kroke et al. in Germany found an increasing BMI z-score to be associated with underestimation [15].

### 4.2. Strengths and Limitations

This study had a long follow-up, included a large sample size and was comprised of participants from a variety of socio-cultural backgrounds living in Amsterdam. Height and weight were measured according to a standardized protocol by trained personnel, and the longitudinal data increased validity for studying a causal relationship. We were able to include multiple explanatory demographic variables. Moreover, enough children with a normal weight participated in this study to also measure the association between maternal perception and child’s weight development among them. For prevention efforts, this is relevant because mothers that underestimate their children’s normal weight may contribute to unhealthy weight gain. The association found in the univariate analyses among children with and without overweight seems partly biased by differences in baseline body weight between the children that were accurately perceived vs. underestimated. Overall, the underestimated group had a lower baseline SDS BMI within the spectrum of normal and overweight, which may explain why mothers were more likely to underestimate their child’s weight. This emphasizes the importance of adjusting for baseline SDS BMI when studying this association. However, the study had limitations that need consideration before drawing conclusions. Firstly, not all recruited mother–child pairs participated in the study at age 5–6 years; thus, selection bias may have affected the observed associations. Secondly, maternal and paternal BMI was self-reported in questionnaires; because participants tend to overestimate their height and weight, this could result in an underestimation of BMI [33]. However, as the present study used BMI as a continuous variable, this has little effect [33]. Thirdly, there was a relatively small sample size of children with overweight and obesity, particularly accurately estimated children with overweight, compared to children with normal weight. This might have influenced the statistical power and explained the loss of significance of the association between maternal perception and child development in children with overweight in the multivariate models. Fourthly, our sample population had unequally divided ethnic groups, with the majority being ethnically Dutch. Nonetheless, the groups were still large enough to show ethnic differences in weight development. Finally, our study has limited generalisability since the sample population consisted of parents and children from a multi-ethnic city of the Netherlands only. We recommend future studies to include larger sample sizes with enough children in all weight groups and to compare other developing and non-developing countries to further explore the role of socio-cultural differences.

### 4.3. Recommendations for Practice

The results indicate that a reduced number of healthcare check-ups after age two, might result in unhealthy weight gain if parents remain unaware of their underestimation of their child’s weight.

Due to the increased expenses that come with more frequent check-ups for children, a more practical recommendation may include tailored feedback by youth healthcare providers or extra check-ups only in those with increased risk of obesity. More specifically, youth healthcare workers may first ask what parents think of their child’s weight at age five, and according to their perceptions of their child’s weight, give personalized advice regarding weight management. The tailored check-ups may enable youth healthcare providers to help parents acquire realistic weight perceptions of their children, which seems important for a long-term preventive approach to overweight and obesity. Our results also suggest that the tailored feedback should be given in a socio-culturally sensitive way in order to reach at-risk groups such as lower educated parents and parents with different ethnic backgrounds.

## 5. Conclusions

In conclusion, maternal underestimation of child’s weight is associated with increased weight development in children with normal weight. A similar trend was seen among children with overweight. Therefore, it seems recommendable that future overweight prevention interventions help parents to acquire a realistic perception of their child’s weight status, including if the child’s weight is normal. Since maternal educational level and ethnicity were important characteristics that were related to maternal perception of child’s body weight and child’s weight development, special attention is needed for socio-cultural differences in future research and practice in preventing childhood overweight.

## Figures and Tables

**Table 1 ijerph-17-05197-t001:** Characteristics of parents and children (mean ± SD/%) from the Amsterdam Born Children and their Development (ABCD) Study (the Netherlands) (*n* = 2189) by weight status and maternal perception of child’s weight at age 5/6.

	Normal Weight Accurate Estimation (*n* = 1872)	Normal Weight Underestimation (*n* = 110)	Overweight Accurate Estimation (*n* = 42)	Overweight Underestimation (*n* = 165)	Total (*n* = 2189)
Maternal age (years) ^c,f^	32.3 (±4.4)	30.5 (±5.4)	31.7 (±4.8)	29.5 (±5.7)	32.0 (±4.7)
Maternal BMI ^c,f^	23.5 (±4.2)	24.4 (±4.5)	26.4 (±5.2)	26.9 (±5.2)	23.9 (±4.4)
Paternal BMI ^a,e^	24.9 (±3.0)	25.8 (±4.9)	26.3 (±3.3)	26.5 (±3.5)	25.0 (±3.2)
Maternal education level ^c f^					
% Low	11.0	32.7	32.5	40.5	14.6
% Middle	19.5	19.6	25.0	23.4	19.9
% High	69.5	47.7	42.5	36.1	65.5
Child age (years)					
5-year-olds	5.7 (±4.2)	5.8 (±0.6)	5.8 (±0.5)	5.9 (±0.5)	5.7 (±0.5)
11-year-olds	11.1 (±0.7)	10.9 (±0.7)	10.9 (±0.6)	10.9 (±0.7)	10.6 (±0.4)
Child sex ^b^					
% Girl	49.8	41.8	28.6	56.4	50.3
% Boy	50.2	58.2	71.4	43.6	49.7
Ethnicity ^c,f^					
% NL	69.3	45.5	50.0	37.0	65.3
% SUR	3.9	8.2	9.5	10.3	4.7
% TUR	2.7	9.1	4.8	11.5	3.7
% MOR	5.2	11.8	14.3	18.2	6.7
% OTH	18.9	25.5	21.4	23.0	19.6
Child BMI (kg/m2)					
5-year-olds ^c,f^	15.5 (±0.9)	14.8 (±0.7)	19.5 (±1.7)	18.7 (±1.4)	15.7 (±1.4)
11-year-olds ^c,f^	17.5 (±2.1)	16.8 (±2.0)	22.8 (±3.8)	22.1 (±3.0)	17.9 (±2.6)
SDS BMI					
5-year-olds ^a,f^	0.1 (±0.6)	−0.4 (±0.5)	2.2 (±0.6)	1.9 (±0.5)	0.2 (±0.8)
11-year-olds ^c,f^	0.0 (±0.9)	−0.3 (±0.9)	1.7 (±0.8)	1.5 (±0.8)	0.1 (±1.0)

Abbreviations: NL, Dutch; SUR, Surinamese; TUR, Turkish; MOR, Moroccan; OTH, Other; BMI, body mass index; SDS, standard deviation scores. ^a, b, c^ indicate significant differences between the normal and overweight group at the level of *p* < 0.05, *p* < 0.01, and *p* < 0.001, respectively. ^d, e, f^ indicate significant differences between accurately estimated and underestimated groups at the level of *p* < 0.05, *p* < 0.01, and *p* < 0.001, respectively.

**Table 2 ijerph-17-05197-t002:** Univariate and Multivariate associations between ‘maternal perception of child’s weight’ and ‘child’s weight development’ (delta SDS BMI) between 5/6 and 11/12 for children with normal weight at age 5/6 (*n* = 1982) and overweight (*n* = 207).

	Normal Weight Children			Children with Overweight		
	Univariate Analysis β (95% CI)	Multivariate Analysis Model 1 ^a^ β (95% CI)	Multivariate Analysis Model 2 ^b^ β (95% CI)	Univariate Analysis β (95% CI)	Multivariate Analysis Model 1 ^a^ β (95% CI)	Multivariate Analysis Model 2 ^b^ β (95% CI)
Maternal Perception						
Accurate estimation	Reference	Reference	Reference	Reference	Reference	Reference
Underestimation	0.19 (0.05, 0.33) **	0.15 (0.01, 0.30) *	−0.11 (−0.28, 0.05)	0.22 (0.01, 0.44) *	0.18 (−0.04, 0.41)	0.06 (−0.20, 0.31)
Child sex						
% Girl		Reference	Reference		Reference	
% Boy		0.13 (0.06, 0.19) ***	0.15 (0.09, 0.22) ***		−0.16 (−0.35, 0.03)	
Baseline SDS BMI	-	−0.06 (−0.11, −0.003) *	−0.17 (−0.23, −0.11) ***	-	−0.19 (−0.35, −0.02) *	−0.34 (−0.56, −0.12) **
Maternal BMI	-	-	0.03 (0.02, 0.04) ***	-	-	0.05 (0.03, 0.07) ***
Paternal BMI	-	-	0.03 (0.02, 0.04) ***	-	-	0.01 (−0.02, 0.04)
Maternal Age	-	-	−0.00 (−0.01, 0.005)	-	-	−0.01 (−0.03, 0.01)
Ethnicity	-	-		-	-	
NL			Reference			Reference
SUR	-	-	0.13 (−0.09, 0.35)	-	-	0.11 (−0.32, 0.55)
TUR	-	-	0.32 (0.11, 0.53) **	-	-	0.13 (−0.22, 0.49)
MOR	-	-	−0.16 (−0.34, 0.02)	-	-	0.04 (−0.29, 0.36)
OTH		-	0.14 (0.05, 0.23) **	-	-	0.16 (−0.1, 0.43)
Maternaleducation level						
Low	-	-	0.23 (0.14, 0.32) ***	-	-	0.13 (−0.16, 0.41)
Middle	-	-	0.25 (0.12, 0.39) ***	-	-	0.22 (−0.07, 0.50)
High	-	-	Reference	-	-	Reference

Abbreviations: NL, Dutch; SUR, Surinamese; TUR, Turkish; MOR, Moroccan; OTH, Other; BMI, body mass index; SDS, standard deviations scores. * *p* ≤ 0.05, ** *p* ≤ 0.01, *** *p* ≤ 0.001. ^a^ Model 1, adjusted for: child sex, baseline SDS BMI. ^b^ Model 2, adjusted for: child sex, baseline SDS BMI, maternal BMI, paternal BMI, maternal age, ethnicity, maternal education level.

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
