# Peer review of "Maternal Underestimation of Child’s Weight at Pre-School Age and Weight Development between Age 5 and 12 Years: The ABCD-Study"

_ijerph, 2020, doi:10.3390/ijerph17145197_

Round 1
Reviewer 1 Report
This study contains significant contents.
However, I think some revisions are needed.
First, P-values are needed in the Table 2. (To see the significance level)
Second, in case of normal weight, the sinificance was disappeared in model 2.
In case of overweight, the significances were diappeared in model 1 and 2.
However, the authors did not fully discussed the reasons.
Third, the more detailed explanation about SDS was needed.
I could not understant the exact meaning of baseline SDS BMI.
Reviewer 2 Report
- In this manuscript I missed a discussion on the cultural and genetic aspects of body perception with regards to your results on Dutch children non-Dutch children.
- Since you have large sample and similar number of girls and boys, a separation would be interesting. In general, it would be interesting to see your results presented separately for boys and girls. That is, when separate analysis for boys and girls would be performed. Also, a discussion on the differences in body composition (during pubertal maturation), physical activity, eating habits relation to parents etc. for boys and girls. How results will change if you do separate analysis for girls and boys?
- Lines 114-119: “Maternal perception was measured through questionnaires in which mothers were asked what they thought of their child’s weight status based on a 5-point scale (1=way too heavy; 2=too heavy; 3=just right; 4=too light; 5=way too light). If mothers perceived their child to be a lower weight category than they actually were (e.g., a child with overweight being perceived as normal weight) we called this On the other hand, if mothers perceived their child’s weight accurately, in the correct weight group, we called this accurate estimation(13).”
- What did you do with children if mothers perceived their child to be a higher weight category than they actually were (overestimation)? How many were there? Why you did not report those?
- Minor comment: Line 84: Academic Medical Center Medical Ethic Committee need to be corrected “Academic Medical Center Medical Ethics Committee”.
Round 2
Reviewer 1 Report
Thank you for appropriate revision.